# DTC-YOLO: Multimodal Object Detection via Depth-Texture Coupling and Dynamic Gating Optimization

**DOI:** 10.3390/s25185731

**Published:** 2025-09-14

**Authors:** Wei Xu, Xiaodong Du, Ruochen Li, Lei Xing

**Affiliations:** School of Transportation, Shandong University of Science and Technology, Qingdao 266590, China; xuwei972@163.com (W.X.); 202383160020@sdust.edu.cn (X.D.); 202282160014@sdust.edu.cn (R.L.)

**Keywords:** multimodal, rgb-lidar, feature-focused fusion, depth-color mapping

## Abstract

To address the inherent limitations of single-modality sensors constrained by physical properties and data modalities, we propose DTC-YOLO (Depth-Texture Coupling Mechanism YOLO), a depth-texture coupled multimodal detection framework. The main contributions are as follows: RGB-LiDAR (RGB-Light Detection and Ranging) Fusion: We propose a depth-color mapping and weighted fusion strategy to effectively integrate depth and texture features. ADF^3^-Net (Adaptive Dimension-aware Focused Fusion Network): A feature fusion network with hierarchical perception, channel decoupling, and spatial adaptation. A dynamic gated fusion mechanism enables adaptive weighting across multidimensional features, thereby enhancing depth-texture representation. Adown Module: A dual-path adaptive downsampling module that separates high-frequency details from low-frequency semantics, reducing GFLOPs (Giga Floating-point Operations Per Second) by 10.53% while maintaining detection performance. DTC-YOLO achieves substantial improvements over the baseline: +3.50% mAP50, +3.40% mAP50-95, and +3.46% precision. Moreover, it maintains moderate improvements for medium-scale objects while significantly enhancing detection of extremely large and small objects, effectively mitigating the scale-related accuracy discrepancies of vision-only models in complex traffic environments.

## 1. Introduction

Driven by consumption upgrading and industrial transformation, the automotive industry is gradually moving toward electrification and intelligence, making autonomous driving a global research hotspot. Autonomous driving can assist or even replace human drivers in performing driving tasks and plays a crucial role in improving traffic conditions, increasing road efficiency, and reducing traffic accidents. Major countries have incorporated the promotion of autonomous vehicles into their top-level strategic plans and issued relevant regulations to support their development, aiming to seize the lead during the automotive industry’s transformation.

Autonomous driving systems mainly comprise three modules: environmental perception, decision-making and planning, and control execution. Among them, the environmental perception system serves as the foundation, utilizing onboard sensors and vehicle-to-everything facilities to detect traffic objects, recognize lane markings, acquire traffic signals, and update localization. The accuracy and real-time performance of perception directly affect subsequent decision-making and control processes. The object detection module, as a core component of perception, requires high precision and robustness in complex and dynamic environments [1].

Most existing mainstream methods rely on single-modal sensors, particularly vision-based detection. However, single-modal approaches face significant challenges under adverse weather conditions such as fog, low light, and snow [2,3,4,5]. While RGB cameras provide rich texture, detail, and color information, they are highly sensitive to lighting conditions, leading to degraded performance in low-light or strongly reflective scenarios [6,7,8]. Moreover, vision-based single-modal detection struggles with multi-scale object recognition in complex traffic environments: models often find it difficult to maintain stable performance for medium-scale targets while simultaneously achieving high detection accuracy for very large objects (e.g., large vehicles) and very small objects (e.g., pedestrians and cyclists) [9,10].

In addition, infrared sensors, millimeter-wave radar, and LiDAR offer certain advantages but also suffer from limitations such as insufficient resolution, missing details, or lack of texture. These physical and modal constraints limit the effectiveness of single-modal enhancement methods and make it difficult to overcome inherent bottlenecks [11].

Therefore, fusing multi-source heterogeneous data and building a cross-modal collaborative perception framework has become a key approach to improving detection performance and reliability in autonomous driving. Studies have shown that heterogeneous modalities formed by RGB and LiDAR exhibit significant complementary advantages: the former provides rich semantic texture, while the latter ensures geometric accuracy [12]. In recent years, various fusion schemes have been proposed. For example, the E2E-MFD (end-to-end synchronous multimodal fusion detection) method [13] integrates visible–infrared fusion with detection into an end-to-end framework, demonstrating excellent performance under adverse weather conditions; Wang et al. [14] proposed a multi-scale cross-modal feature interaction mechanism, achieving breakthroughs in image-point cloud alignment but with high computational cost; Xu et al. [15] designed a LiDAR-RGB dual-branch architecture that realizes information complementarity through late fusion, yet lacks dynamic adaptability. Overall, existing studies still face limitations in semantic alignment, computational efficiency, and scale robustness [16,17,18,19].

Against this background, this paper proposes a cross-modal fusion detection framework that effectively integrates LiDAR point clouds and RGB images to achieve robust detection of objects at different scales in complex traffic environments. The main contributions of this study are as follows:We propose a depth-color mapping and weighted fusion strategy to effectively integrate depth and texture features.A hierarchical perception-channel decoupling-spatial adaptive fusion mechanism is proposed to mitigate depth artifacts caused by point cloud projection and enhance the complementarity between RGB and LiDAR features.An asymmetric dual-path downsampling module is designed to reduce computational complexity while maintaining detection accuracy, providing a lightweight solution for real-time multimodal detection.

Experiments on the KITTI dataset demonstrate that the improved algorithm achieves strong performance across objects of different scales in complex traffic scenarios: while maintaining stable gains for medium-scale targets, it significantly improves detection accuracy for very large and very small targets (e.g., pedestrians and cyclists), showing superior robustness and application potential compared with existing methods.

This article is organized as follows: Section 1 introduces the background, research motivation, and outlines our cross-modal detection framework and contributions. Section 2 reviews related work on single-modality and multimodal object detection, critically analyzing the strengths and limitations of existing methods. Section 3 details the proposed improvements in DTC-YOLO, focusing on multi-scale detection under complex traffic and adverse weather conditions. Section 4 presents comprehensive comparisons with state-of-the-art single- and multimodal detectors. Section 5 concludes the paper by summarizing the key innovations, discussing limitations, and outlining future research directions.

## 2. Related Work

In 2012, Krizhevsky et al. [20] proposed Convolutional Neural Networks (CNNs), marking a significant advancement in object detection as a crucial application domain of computer vision. The field gradually evolved into two categories of object detection algorithms: single-stage and two-stage approaches.

Two-stage algorithms primarily involve two steps: (1) region proposal generation and (2) classification/regression [21]. In 2014, Girshick et al. [22] pioneered the R-CNN model, which employed selective search for region proposals, CNN for feature extraction, SVM for classification, and regression for bounding box refinement. Subsequent developments included Fast R-CNN (2015) [23], which introduced RoI Pooling, shared convolutional features, and joint optimization of classification/regression, and Faster R-CNN (2016) [24], which proposed Region Proposal Networks (RPNs) for end-to-end training and significantly improved speed. While achieving high detection accuracy (particularly in complex scenes and small object detection), these two-stage algorithms suffer from high computational costs and poor real-time performance (e.g., Faster R-CNN operates at ∼5 FPS).

Single-stage algorithms emerged to fundamentally address these limitations. Current mainstream detectors, notably the YOLO series, predominantly use CSPNet or ELAN variants as computational backbones [25]. These models directly output class labels and bounding boxes through a single network without region proposal generation, resulting in lightweight architectures and high-speed operation suitable for real-time applications (e.g., YOLOv5 achieves 140 FPS). Recent years have also seen other detectors like SSD [26] (which underperforms in multi-scale detection of small objects), EfficientDet [27] (achieving efficiency through compound scaling but requiring complex parameter tuning and showing limited task adaptability), and Transformer-based detectors (e.g., DETR [28], Deformable DETR [29]). However, DETR-series detectors exhibit slow convergence, high computational demands, and challenges in real-time deployment and domain adaptation without pretrained models. Consequently, YOLO-series detectors remain the most widely adopted solutions for real-time object detection.

In recent years, single-stage object detection models have undergone continuous optimization and refinement, with their detection accuracy gradually surpassing that of two-stage methods [30]. In 2024, Wan et al. [31] proposed the YOLO-MIF detection model, which enhances channel information by generating pseudo multi-channel grayscale images to mitigate noise and blurring issues, significantly improving detection accuracy. This approach outperforms Faster R-CNN by 4.8%, demonstrating superior balance between precision and efficiency. Separately, Wang et al. [32] introduced an advanced Gather-and-Distribute (GD) mechanism that integrates convolutional and self-attention operations to strengthen multi-scale feature fusion capability. This mechanism achieves an optimal trade-off between latency and accuracy across all model scales, effectively addressing information fusion challenges in YOLO-series models. Concurrently, Chen et al. [33] developed YOLO-MS, a highly efficient and robust detector. By analyzing the impact of multi-branch features in base blocks and varying convolution kernel sizes on multi-scale detection performance, they proposed a novel strategy to enhance multi-scale feature representation in real-time detectors. This strategy further serves as a plug-and-play module to improve the performance of other YOLO variants.

### Multi-Source Data Fusion

With the gradual maturation of vision-based object detection algorithms, the performance improvement achievable through single-modal enhancement methods has become increasingly limited due to the inherent physical constraints and data modality restrictions of single sensors. Consequently, growing research efforts have shifted toward constructing cross-modal collaborative perception frameworks to enhance detection performance via multi-source heterogeneous data fusion [34].

Long et al. [35] proposed an unsupervised image fusion model, RXDNFuse, based on aggregated residual dense networks, achieving synergistic retention of infrared thermal radiation and visible light textures. However, in infrared-RGB fusion, nonlinear interference between thermal radiation properties and visible spectra (e.g., color distortion in high-temperature regions, over-smoothing of visible textures) persists, particularly under complex illumination where low signal-to-noise areas lack adaptive thermal radiation weighting mechanisms, leading to blurred edges of thermal targets in weak-texture backgrounds. Additionally, spatial resolution discrepancies between infrared sensors and RGB cameras may induce cross-modal feature misalignment (e.g., pixel-level misalignment between small-scale heat sources and RGB textures), while the absence of temporal consistency constraints in dynamic scenes could exacerbate motion artifacts.

Li et al. [36] introduced an illumination-adaptive multispectral pedestrian detection framework (IAF R-CNN), which enhances robustness under varying lighting conditions through dynamic weighted fusion of visible and thermal imaging modal detection confidences, achieving state-of-the-art performance on the KAIST dataset. However, their weighting mechanism heavily relies on precise illumination condition measurements (risking error propagation in extreme low-light or glare scenarios), and the computational overhead of multi-modal feature fusion remains underexplored, potentially hindering real-time requirements for lightweight deployment (e.g., onboard edge devices). Furthermore, validation was limited to the KAIST dataset, leaving generalization to other complex scenarios (e.g., rain/fog interference, dense occlusions) unaddressed.

Wang et al. [14] developed a multi-scale cross-modal feature interaction mechanism that enhances feature discriminability through pixel-point cloud interaction fusion and attention-based weighting. However, their dense cross-modal feature interactions (e.g., multi-level bidirectional information transfer) and dynamic weight computations (via attention mechanisms) may introduce significant computational overhead. Asvadi et al. [1] proposed a 3D-LiDAR (depth/reflectance maps) and RGB camera-based multi-modal vehicle detection system, improving detection accuracy via ANN-based late fusion. Nevertheless, their late fusion strategy operates solely at the bounding box level, failing to fully exploit early/mid-level cross-modal feature interactions. Additionally, LiDAR data upsampling for dense depth maps and high-resolution reflectance map processing may impose computational bottlenecks for real-time edge deployment.

Xiao et al. [37] applied RGB-depth fusion to end-to-end driving systems, evaluating the impact of early, mid-, and late fusion strategies on autonomous driving performance through CARLA simulator experiments with conditional imitation learning (CIL). Results demonstrated that early fusion of multimodal perception data (RGB + depth) outperforms single-modal approaches. Limitations include: (1) exclusive reliance on CARLA simulations, which may diverge from real-world deployment scenarios, and (2) unaddressed challenges in handling sensor noise while ensuring model robustness.

This section reviews advances and technical bottlenecks in real-time object detection and multimodal fusion. Object detection has evolved from two-stage paradigms (e.g., Faster R-CNN) to single-stage architectures (e.g., YOLO series), achieving coordinated optimization of accuracy and efficiency. Single-stage models, leveraging end-to-end architectures and lightweight designs, dominate real-time detection. However, existing algorithms remain constrained by the information entropy boundaries of single-modal data, particularly in open-set conditions (e.g., low visibility, adverse lighting conditions, and occlusions), which exacerbate feature confusion and false detection rates.

Therefore, the core proposition of this study is to achieve efficient fusion of LiDAR and RGB multi-source heterogeneous data and make up for the detection bottleneck of vision-based target detection algorithms in bad weather.

## 3. Method

### 3.1. Depth–Texture Coupling-Based Feature Enhancement Strategy

The limitations of single-modality enhancement methods in improving detection performance have become increasingly apparent due to the intrinsic differences in sensor characteristics and data modalities [2]. To address this issue, we propose a multimodal perception strategy based on depth-texture coupling (DTC). This framework improves the detection performance of targets of different scales by implementing weighted fusion of LiDAR and RGB images.

#### 3.1.1. Spatiotemporal Alignment of Heterogeneous Sensors

To construct a multimodal joint calibration model for LiDAR and camera, we first consider the imaging process. When the effects of camera perspective distortion are neglected, the imaging process can be treated as a linear operation. According to Zhang Zhengyou’s [38] calibration method, the camera imaging model can be approximated by a pinhole model. This model involves several coordinate systems, including the radar, camera, pixel, and image coordinate systems. The relationships between these coordinate systems are illustrated in Figure 1.

The intrinsic matrix maps 3D camera coordinates to 2D pixel coordinates, while the extrinsic matrix defines the rigid transformation between LiDAR and camera coordinates. The pixel coordinate system (measured in pixels) specifies precise 2D positions within the image.

(1) Projection Matrix Construction (as formulated in Equation (Equation 1)).(1)$$P=\underset{K}{\underset{︸}{\left[\begin{array}{cccc} f_{x} & 0 & c_{x} & 0\\ 0 & f_{y} & c_{y} & 0\\ 0 & 0 & 1 & 0 \end{array}\right]}}\cdot \underset{\left[R|T\right]}{\underset{︸}{\left(\begin{array}{cc} R_{v}^{c} & T_{v}^{c}\\ 0_{1\times 3} & 1 \end{array}\right)}}$$
where [R∣T] denotes the camera extrinsic matrix, transforming points from the LiDAR coordinate system to the camera coordinate system; $R_{v}^{c}\in R^{3\times 3}$ represents the rotation matrix; $T_{v}^{c}\in R^{3\times 1}$ is the translation vector; *K* corresponds to the camera intrinsic matrix, projecting points from the camera coordinate system to the pixel coordinate system; $f_{x}$ and $f_{y}$ denote focal lengths; and $c_{x}$ and $c_{y}$ define the principal point coordinates.

(2) Three-Dimensional Point Cloud Projection

The projection process is formulated as shown in Equations ([Disp-formula FD2a-sensors-25-05731]) and ([Disp-formula FD2b-sensors-25-05731]):(2a)$$\begin{array}{cc} \left[\begin{array}{c} u\\ v\\ 1 \end{array}\right]\; & =\frac{1}{z}\cdot K\cdot \left[R|T\right]\cdot P_{v} \end{array}$$(2b)$$\begin{array}{cc} P_{v} & ={\left[\begin{array}{c} X_{v}\\ Y_{v}\\ Z_{v}\\ 1 \end{array}\right]}^{T} \end{array}$$
where $P_{v}$ represents the homogeneous coordinates of a point in the LiDAR coordinate system; *u* and *v* denote the normalized pixel coordinates; and *z* is the depth in the camera coordinate system.

#### 3.1.2. RGB-LiDAR Cross-Modal Fusion to Achieve Self-Attention Focus

In the fields of autonomous driving and robotic perception, failures in single-modal systems often compromise overall system robustness. RGB images and LiDAR point clouds exhibit intrinsically complementary characteristics. RGB images provide rich textural details and high-dimensional semantic representations but lack depth information and are susceptible to illumination variations. Conversely, LiDAR delivers precise geometric structure information with illumination invariance, yet its spatial sparsity limits the availability of semantic context. We propose a cross-modal encoder built upon cross-modal feature alignment and depth-color mapping. The workflow is illustrated in Figure 2.

To address depth scale variations across different scenes, we establish a dynamic normalization mechanism to ensure visual consistency of the heat maps. By applying a Jet colormap function, the normalized depth values are encoded into a [0, 255] pseudo-color space, producing pseudo-depth images that preserve depth fidelity while enhancing visual distinction. Furthermore, a weight adjustment mechanism is incorporated during pixel-wise fusion with the original RGB image. This mechanism ensures that the resulting depth-texture fused images achieve high-fidelity depth representation while maintaining high visual quality and adaptability for subsequent processing, as specified in Equation (Equation 3).(3)$$I_{\mathrm{fuse}\;}=\alpha \cdot I_{rgb}+\left(1-\alpha \right)\cdot CM_{\mathrm{jet}\;}\left(\left\lfloor 255\cdot \frac{d_{i}}{D_{\max}}\right\rfloor \right)$$
where $d_{i}$ is the point cloud depth value, $D_{\max}$ is the maximum depth of the scene, and $CM_{jet}$ is the Jet color gamut mapping function. Jet color mapping is used to convert the single-channel depth map into an RGB heat map, mapping the minimum value of the depth map to blue, the maximum value to red, and intermediate values to cyan, yellow, etc. $I_{rgb}$ denotes the corresponding RGB image pixel, and α is the fusion weight, set to 0.6 in this work.

Particularly under low-light, rainy, or foggy conditions, where target textures are severely degraded, the depth heatmap compensates for texture-weak regions. Moreover, the heatmap distribution inherently induces a self-attention mechanism, guiding the network to focus on critical regions. As demonstrated in Figure 3a,b, after integrating DTC, both the proportion of high-activation pixels in the receptive field (where color depth exceeds a threshold) and the peak activation intensity increase, indicating that DTC’s self-attention mechanism enhances focus on crucial local features, such as wheel edges and headlight structures. Simultaneously, the receptive field expands, significantly enlarging the scope of contextual perception. This facilitates the fusion of multi-scale global information in complex scenes.

The cross-modal encoder, constructed via cross-modal feature alignment and depth-color mapping, effectively exploits the complementary characteristics of RGB-LiDAR data, achieving a coupled representation of geometric depth and textural details within a unified feature space. Experiments (Figure 4) demonstrate that this design significantly mitigates the vulnerability of single-modal systems to disturbances such as occlusion and low illumination.

### 3.2. Adaptive Dimension-Aware Focused Fusion Network

Traditional YOLO employs fixed-weight feature concatenation and a uniform fusion strategy across all regions, making it insensitive to the high-dimensional semantic representations of depth-texture coupled feature maps. This study proposes an Adaptive Dimension-aware Focused Fusion Network (ADF^3^-Net). This network achieves dynamic selection and optimized integration of multi-scale information by establishing a feature fusion paradigm of hierarchical perception, channel decoupling and spatial adaptation. Through the dynamic gating mechanism, the features of different dimensions are adaptively integrated, which significantly improves the multi-scale modeling capability of the model while maintaining the computational efficiency. The core processing flow is shown in Figure 5.

#### 3.2.1. Multi-Scale Collaborative Feature Encoding

Input feature maps are processed through differentiated transformations in the ADF^3^-Net parallel network to capture multi-granularity spatial information:

(1) Shallow detail enhancement:

Shallow features contain rich details, including edges and textures but also suffer from significant noise. Conventional downsampling often causes spatial detail loss and limits the receptive field. In contrast, controllable sparse sampling combined with dilated convolution downsampling expands the effective receptive field to K+(k−1)(d−1) (with k=3 and d=2) without increasing the number of parameters. This captures long-range dependencies and improves detection performance for targets of various scales. Channels are expanded while resolution is reduced, preserving high-frequency details (Equation (Equation 4)).(4)$$F_{high}^{\prime }=C_{3\times 3}^{d=2,s=2}\left(F_{high}\right)\in R^{\left(C\times H/2\times W/2\right)}$$
where d=2 denotes the dilation rate, S=2 represents the stride, and $R^{\left(C\times H/2\times W/2\right)}$ specifies the output feature map dimensions as C×H/2×W/2.

(2) Mid-level Feature Reconstruction

Perform channel transformation on mid-level features. To enhance feature fusion, we employ an edge-aware attention mechanism that generates spatially adaptive attention maps from these features. These maps enable dynamic weighted fusion between shallow detail features (high-frequency components) and deep semantic features (low-frequency components). The process is shown in Equation (Equation 5):(5)$$F_{mid}^{\prime }=C_{1\times 1}\left(F_{mid}\right)\in R^{\left(C\times H/2\times W/2\right)}$$
where $C_{1\times 1}$ denotes the 1×1 convolution operation for channel adjustment.

(3) Deep Semantic Expansion

The deep features are first adjusted by 1×1 convolution, and then the spatial resolution is reconstructed by upsampling using a bilinear interpolation function. This avoids high-frequency noise interference while suppressing the aliasing effect, as shown in Equation (Equation 6):(6)$$F_{\mathrm{low}\;}^{\prime }=ℬ\left(C_{1\times 1}^{C^{\prime }\to C}\left(F_{\mathrm{low}\;}\right)\right)\in R^{\left(C\times H/2\times W/2\right)}$$
where ℬ(·) denotes the bilinear interpolation operator, ensuring alignment of spatial dimensions with mid-level features.

#### 3.2.2. Channel Decoupling and Adaptive Feature Fusion

(1) Channel Decoupling

The high-, medium-, and low-level features output by the above process are further decoupled from the feature space through tensor block operations to capture multi-granular features, as shown Equation (7):(7a)$$\begin{array}{cc} F_{\mathrm{high}\;}^{\prime } & ={⊕}_{k=1}^{4}F_{\mathrm{high}\;}^{\left(k\right)}, \end{array}$$(7b)$$\begin{array}{cc} F_{\mathrm{mid}\;}^{\prime } & ={⊕}_{k=1}^{4}F_{\mathrm{mid}\;}^{\left(k\right)}, \end{array}$$(7c)$$\begin{array}{cc} F_{\mathrm{low}\;}^{\prime } & ={⊕}_{k=1}^{4}F_{\mathrm{low}\;}^{\left(k\right)} \end{array}$$
where ⊕ denotes the channel-wise split-and-concatenation operation, generating four sub-feature groups ${\left\{F^{\left(k\right)}\right\}}_{i=1}^{4}$ for each frequency band.

(2) Adaptive Feature Fusion

In the proposed mechanism, an attention mask *d* is generated from intermediate features. This mask dynamically adjusts the contribution weights of shallow details *p* and high-level semantics *i*. Traditional attention mechanisms compute similarities for all positions (e.g., the $QK^{T}$ matrix), resulting in $O(N^{2})$ complexity, where *N* denotes the spatial size of the feature map (H × W). In contrast, this mechanism uses only mid-level features to generate a gating mask *d*, followed by a Sigmoid function, element-wise Hadamard multiplication, and addition, all element-wise operations with O(N) complexity, i.e., linear with respect to the feature map size rather than quadratic. This avoids the high $O(N^{2})$ overhead while preserving spatial adaptability and significantly improving computational efficiency. The specific formulation of this process is shown in Equation (Equation 8). (8)$$\begin{array}{c} 𝒢(p,i,d)=\sigma (d)⊙p+(1-\sigma (d))⊙i \end{array}$$
where σ(·) denotes the Sigmoid function, and ⊙ represents the Hadamard product.

#### 3.2.3. Subspace Reorganization and Residual Enhancement

Each sub-feature group is independently fused across layers, and the channel recalibration operator is used to reorganize and distill the channels, extract key feature information, and reduce mutual interference. This pipeline is formalized in Equation (Equation 9):(9)$$\begin{array}{c} F_{\mathrm{reorg}\;}=ℛ\left(\underset{k=1}{⊕}F_{\mathrm{fuse}\;}^{\left(k\right)}\right) \end{array}$$
where ⊕ denotes channel-wise concatenation, $F_{\mathrm{fuse}}$ is the local feature representation obtained by fusing low-level, middle-level, and high-level features of each block sub-channel, and ℛ(·) is a 1×1 convolution-based channel recalibrator.

To mitigate information decay in feature pyramids, a residual enhancement module is incorporated, as formulated in Equation (Equation 10). It performs residual enhancement by integrating the original feature map to avoid information loss.(10)$$\begin{array}{c} F_{\mathrm{out}\;}=ℬ𝒩\left(F_{\mathrm{reorg}\;}+F_{\mathrm{skip}\;}\right) \end{array}$$

Here, $F_{\mathrm{skip}\;}$ represents skip connections from channel-projected original features, and ℬ𝒩 is batch normalization.

Overall, the ADF^3^-Net module effectively enhances feature integration by enabling multi-dimensional perception fusion across spatial, channel, and semantic dimensions. This approach not only boosts the model’s ability to capture both high-frequency edge details and low-frequency information but also enhances feature collaboration. As shown in Figure 3, the activation area and intensity of the receptive field are significantly increased.

Unlike traditional global attention mechanisms, ADF^3^-Net generates attention masks from intermediate features, achieving more precise feature fusion while improving computational efficiency. Additionally, residual connections mitigate information loss when features are passed across pyramid levels, ensuring accuracy and reliability. Overall, while preserving computational efficiency, ADF^3^-Net significantly strengthens the model’s multi-scale modeling capabilities.

### 3.3. Lightweight Dual-Branch Heterogeneous Downsampling Network

To further reduce computational costs, this section proposes ADown, a high-efficiency downsampling module based on dual-path heterogeneous feature extraction, as illustrated in Figure 6. The module achieves multi-modal feature fusion through parallel heterogeneous convolution operations, significantly enhancing feature representational capacity while maintaining computational efficiency. Specifically, ADown employs a dual-branch topology to differentially process input features:

#### 3.3.1. Dual-Branch Heterogeneous Feature Extraction

Unlike traditional single-path downsampling methods, this module first applies a 2×2 average pooling (stride = 1, padding = 0) for preliminary feature smoothing and noise suppression. The resulting feature map is then split along the channel dimension into two independent subspaces:

Branch 1: A 3×3 convolution (stride = 2) performs spatial downsampling to capture localized detail features.

Branch 2: A 3×3 max pooling (stride = 2) enhances salient region responses, followed by a 1×1 convolution for channel dimension transformation.

This heterogeneous processing strategy allows the output features to simultaneously incorporate smooth global statistical properties (from average pooling) and localized salient features (from max pooling), forming complementary representations, as formalized in Equation (11).(11a)$$\begin{array}{cc} Y_{1} & ={\mathrm{Conv}}_{3\times 3}\left(\mathrm{AvgPool}\left(X\right)\right) \end{array}$$(11b)$$\begin{array}{cc} Y_{2} & ={\mathrm{Conv}}_{1\times 1}\left(\mathrm{MaxPool}\left(X\right)\right) \end{array}$$(11c)$$\begin{array}{cc} Y & =\mathrm{Concat}[Y_{1},Y_{2}] \end{array}$$
where $X\in R^{\left(C\times H\times W\right)}$ denotes the input feature map, and $Y\in R^{\left(C\times H/2\times W/2\right)}$ represents the output feature map. This operation achieves 2× spatial downsampling while constructing a semantically enriched feature space through feature reorganization, thereby providing high-quality input foundations for subsequent feature pyramid networks.

#### 3.3.2. Lightweight Computational Optimization

The proposed ADown module significantly improves computational efficiency compared to traditional stride-2 3×3 convolutions. For an input feature map of dimensions H×W, the theoretical FLOPs of a standard 3×3 convolution can be expressed as $FLOPs=9kC^{2}$ (where k=H×W). In contrast, the ADown module reduces computational complexity to $FLOPs=2.25kC^{2}+0.25kC^{2}=2.5kC^{2}$, corresponding to a 72% reduction in FLOPs. This design significantly reduces model complexity while maintaining equivalent downsampling functionality.

Within the ADown module, channel segmentation is combined with multi-branch pooling operations to realize lightweight heterogeneous downsampling. The core innovation lies in its dual-branch heterogeneous structure, integrating average pooling and max pooling. This design enables effective extraction of multi-scale feature information. By splitting the input feature map and applying pooling operations, the module reduces the portion of features directly involved in convolution, thereby substantially lowering computational load. This approach provides high-quality inputs for subsequent feature pyramid networks while keeping parameter overhead minimal.

### 3.4. DTC-YOLO Network Structure Design

We present a multimodal fusion framework for target detection. Its key innovation is a depth–texture–semantic collaborative representation system, enabling cross-modal feature optimization through a multi-path heterogeneous processing mechanism. As shown in Figure 7, the framework consists of three key modules:

(1) Depth-Texture Dual-Stream Encoder: Handles RGB-LiDAR heterogeneous data, achieving cross-modal feature alignment and constructing a joint depth-texture representation space.

(2) Lightweight Dual-Branch Feature Extraction Network: Employs the ADown module to replace the 3rd, 5th, and 7th convolutional layers, reducing model complexity while enhancing feature extraction quality.

(3) Multi-Scale Feature Fusion Network (ADF^3^-Net): Processes features from multiple layers, decouples frequency information through tensor block operations, and applies adaptive weighted fusion, improving feature collaboration and multi-scale target detection performance.

We quantitatively evaluated the model’s feature perception by visualizing the receptive field distribution of the ninth neural layer, as shown in Figure 3. The results indicate that the improved DTC-YOLO surpasses the baseline in three key aspects of receptive field characteristics:

(1) Expanded Receptive Field Area: The receptive field is significantly enlarged, indicating enhanced contextual perception, which facilitates the integration of multi-scale global information in complex scenes.

(2) Increased Activation Intensity: The proportion of pixels in highly activated regions has increased, accompanied by a rise in peak activation intensity. This demonstrates that the network more effectively focuses on critical local features, such as wheel edges and headlight structures.

(3) Dual-Modal Receptive Field Distribution: The receptive field exhibits a “wide coverage–local enhancement” pattern while maintaining uniform spatial coverage, high-intensity response clusters form in target-sensitive areas. This mitigates the traditional trade-off between receptive field expansion and feature response dispersion, providing a robust basis for accurate multi-scale target detection and enabling collaborative optimization of global and local features.

## 4. Experiment and Analysis

### 4.1. Experimental Conditions and Datasets

(1) Experimental Setup

The hardware and software configurations were specified as follows:

Hardware: 13th Gen Intel (R) Core ^TM^ i7-13620H CPU @ 2.40 GHz, NVIDIA GeForce RTX 4060 GPU, 16 GB RAM, Device Manufacturer: Lenovo, Beijing, China; Software: Windows 11 OS, Python 3.9, PyTorch 2.0.0 (GPU-accelerated).

(2) Dataset

The KITTI dataset [39,40] was adopted as the benchmark, a widely recognized multimodal dataset in autonomous driving research. Its spatiotemporally synchronized parameters and coverage of complex road scenes provide an ideal validation environment for point cloud-image cross-modal fusion studies. The dataset contains 7481 annotated frames categorized into nine classes. Six primary traffic participant classes were selected for experimentation:

Car (Truck, Car, Van, Tram): 58.3%; Pedestrian: 28.1%; Cyclist: 13.6%; Notably, 34.7% of objects exhibit partial occlusion, and 17.2% suffer from image truncation (truncation > 0.15), effectively validating algorithm robustness in real-world driving scenarios. Following the official protocol, the dataset was partitioned into training/validation/test sets at a 7:2:1 ratio. Input training images were resized to a resolution of 1216 × 384 pixels. The number of training epochs was set to 100, and both the initial learning rate (lr0) and the final learning rate factor (lrf) were set to 0.01.

To evaluate the proposed DTC-YOLO under low-visibility and weak-lighting conditions, a physics-based data augmentation strategy was implemented:

The Atmospheric Scattering Model simulated fog-induced visibility degradation, the Rain Streak Synthesis algorithm generated rainfall interference, and the Illumination Remapping technique constructed low-light nighttime scenes. During training, these augmentation methods were randomly applied with a 2:2:3:3 probability distribution across four conditions: foggy (Figure 8a), low-light (Figure 8b), rain (Figure 8c), and clear weather (Figure 8d). This multi-condition training set systematically increased scene complexity, enabling rigorous evaluation of algorithm degradation characteristics and generalization capabilities in extreme environments. This strategy is depicted in Figure 8.

### 4.2. Evaluation Indicators

In the research phase, GFLOPs is usually used to evaluate the computational resource requirements of an algorithm, while precision (P), recall (R), and mean average precision (mAP) are employed to analyze algorithm performance. The mAP50 metric serves as the primary indicator for assessing detection accuracy.

Each detection result can be classified into one of the four scenarios outlined in the confusion matrix (Table 1). These classifications enable the calculation of four key metrics: P, R, mAP50, and mAP50-95. The components of the confusion matrix are defined as follows:

True Positive (TP): Positive samples correctly predicted as positive. True Negative (TN): Negative samples correctly predicted as negative. False Negative (FN): Positive samples erroneously predicted as negative. False Positive (FP): Negative samples erroneously predicted as positive.

The mathematical formulations for the metrics mAP, P, R, and average precision (AP) are defined in Equation (12).(12a)$$\begin{array}{cc} mAP & =\frac{1}{n}\sum_{i=1}^{n}AP_{i} \end{array}$$(12b)$$\begin{array}{cc} P & =\frac{TP}{TP+FP} \end{array}$$(12c)$$\begin{array}{cc} R & =\frac{TP}{TP+FN} \end{array}$$(12d)$$\begin{array}{cc} AP & =\int_{0}^{1}P\left(R\right)dR \end{array}$$

### 4.3. Multi-Object Detection Experiments

#### 4.3.1. Comparative Experiments

In this study, we selected multiple types of mainstream target detection algorithms, including two-stage detectors (Faster RCNN), single-stage detectors (SSD, YOLO series), and detectors based on Transformer architectures (RT-DETR), as benchmark models. Systematic comparative experiments were conducted on the KITTI dataset. As shown in Table 2, the DTC-YOLO model proposed in this paper shows significant advantages in comprehensive detection performance indicators, specifically.

Enhanced Detection Accuracy:

In terms of the mAP50 metric, DTC-YOLO achieves a 1.67% improvement over YOLOv8, one of the high-performance models in the YOLO series, and a 2.82% gain compared with the baseline model. This improvement is primarily attributed to the model’s enhanced detection capability for small-scale objects (i.e., targets with a resolution smaller than 32 × 32 pixels [44]). Moreover, under the more rigorous mAP50-95 evaluation metric, DTC-YOLO achieves a 1.46% performance gain, which may result from the geometric priors introduced by the DTC module, thereby enhancing the model’s bounding box regression for occluded objects.

Architectural Efficiency:

Compared with the Transformer-based RT-DETR model, DTC-YOLO achieves a 4.95% mAP50 improvement while maintaining its computational efficiency advantage. It is verified that the multi-scale adaptive feature fusion mechanism designed in this paper has the potential to be comparable to the attention mechanism in terms of feature expression ability.

Improved Multi-Scale Detection:

The 3.94% improvement in precision indicates the model’s strong capability in suppressing background noise and enhancing boundary localization accuracy. This benefit is attributed to the enhanced multi-scale feature representation and contextual modeling capabilities brought by the depth-texture coupling (DTC) and the Adaptive Dimension-aware Feature Focusing and Fusion Network (ADF^3^-Net).

As shown in Table 3, we compare the performance of DTC-YOLO with other multimodal detectors on the KITTI dataset. Compared with YOLO11+Normal mapping, DTC-YOLO achieves a 1.94% improvement in mAP50 and a significant 7.17% increase in mAP50-95. The advantage is even more pronounced over the Faster RNN-based multimodal model, with improvements of 4.08% in mAP50 and 18.4% in mAP50-95.

Among the Ms-YOLO series of multimodal models, DTC-YOLO also outperforms the best-performing variant, Ms-YOLOm, achieving a 1.32% increase in mAP50 and an 8.73% improvement in the more stringent mAP50-95 metric. It is worth noting that compared to the current top-performing model MM-Net, DTC-YOLO further improves the mAP50-95 by 2.55%. Although there is still a gap in mAP50 between DTC-YOLO and MM-Net, it is important to emphasize that DTC-YOLO is a general-purpose detection model optimized for multi-scale targets, whereas MM-Net is specifically designed for pedestrian detection—a single-class target. This contextual difference is crucial for understanding the comparison of their evaluation metrics.

#### 4.3.2. Ablation Experiment

(1) Depth-Texture Collaboration (DTC) Encoder.

In this section, we present a performance analysis of the depth-texture collaboration encoder, which is pivotal to the proposed DTC-YOLO model. As shown in Table 4, the introduction of the DTC structure leads to a 2.27% increase in the strict IoU metric, surpassing the 1.16% gain in mAP50. This indicates that the integration of depth information significantly enhances target localization accuracy, particularly in scenarios involving occlusion or truncation. Geometric priors derived from point cloud projections effectively reduce bounding box localization errors, thereby improving detection precision.

The 7.81% increase in recall is primarily attributed to the enhanced detection of occluded and small-scale targets. Depth information quantifies the continuity of target surfaces, providing additional geometric cues for occluded regions and distant small-scale targets. This improvement is particularly evident in the detection of the “Person sitting” and “Pedestrian” categories, as illustrated in Figure 9. For instance, the density distribution of point clouds helps generate attention masks, guiding the model to focus on areas with low texture but high depth gradients. This enables the detection of weakly textured targets that are often overlooked by traditional RGB-based methods, such as vehicles at night and pedestrians in shadows (Figure 10a,c,e,g,i,j), thus significantly increasing the number of true positives (TP).

Conversely, the 6.12% decrease in precision is mainly due to false positives introduced by depth noise. During point cloud projection, certain elements, such as vegetation or equipment, may be erroneously assigned pseudo-depth features, leading to “depth ghosting” and false detections (Figure 11a,b). Furthermore, in low-light or reflective scenarios, noise in RGB images combined with LiDAR signal scattering introduces feature ambiguity, which may result in the erroneous accumulation of classification confidence. For instance, reflections on building facades may be misclassified as metallic objects, with the likelihood of false detection further amplified when depth information is incorporated (Figure 11c,d).

(2) ADF^3^-Net Enhanced Backbone.

As shown in Table 4, the integration of the ADF^3^-Net architecture leads to significant improvements in both mAP50 (↑ 2.54%) and mAP50-95 (↑ 2.24%), indicating that the network effectively improves the joint representation of depth and texture features. The network’s dynamic gated fusion mechanism suppresses attention weights in non-target regions to near zero, mitigating the “Depth Ghosting” issue resulting from point cloud projection. Consequently, the number of false positives decreases, leading to a 7.50% improvement in precision.

Additionally, the network’s blocking strategy enhances the feature representation of small targets within specific subspaces, thereby improving detection performance for small-scale objects. As illustrated in Figure 9, the network achieves higher detection accuracy for three types of small targets: Person sitting, Pedestrian, and Car-Cyclist.

Furthermore, the network achieves synergistic optimization, as evidenced by a slight increase in Recall (↑ 0.74%) and a substantial improvement in precision (↑ 7.50%). This indicates that the model not only maintains a high detection rate but also substantially improves detection quality. This improvement can be attributed to the cross-modal resonance mechanism, which amplifies the responses of target keypoints (Figure 3c), thereby improving the model’s noise suppression and feature representation capabilities.

(3) Lightweight Dual-Branch Heterogeneous Downsampling (ADown).

As shown in Table 4, an asymmetric dual-path downsampling module (ADown) is introduced, resulting in a 10.53% reduction in computational cost (GFLOPs) via a high- and low-frequency decoupling design while maintaining high model performance. Additionally, the dual-branch heterogeneous processing design within ADown enables coordinated representation of detailed features (e.g., edges and textures) and global semantic information. This design reduces feature redundancy and allows the classifier to better distinguish foreground from background, as reflected by a 2.52% increase in Precision. The introduction of the ADown module does not change mAP50, suggesting performance saturation under the less stringent detection criterion. This highlights ADown’s core advantage: significantly improving foreground–background separation to reduce feature redundancy while lowering computational cost.

#### 4.3.3. Comprehensive Performance Analysis

(1) Breakthrough improvement in small target detection.

As shown in Figure 9, compared with YOLO11, DTC-YOLO achieves a substantial improvement in the detection performance of extremely small targets: Person sitting (19.61% increase), Pedestrian (5.62% increase), and Car-Cyclist (3.58% increase). By integrating depth data from the 3D point cloud, the visual limitations of single-modal systems are effectively mitigated, significantly enhancing detection performance under low-visibility, complex road conditions, occlusion, and other challenging scenarios. In particular, the missed detection rate for low-texture, small-scale, and blurred targets is substantially reduced, as shown in the comparison between Figure 10a and Figure 10b, Figure 10c and Figure 10d, Figure 10e and Figure 10f, Figure 10g and Figure 10h, and Figure 10i and Figure 10j.

(2) Improved detection performance of extremely large objects.

The framework achieves significant improvements on extremely large (Tram: ↑ 3.18%) and very small objects, while maintaining modest gains on medium-sized objects. This is due to the dynamic weighted fusion mechanism of ADF^3^-Net’s high- and low-frequency features, which improves the model’s detection performance for targets of different scales.

(3) Failure Case Analysis.

Analysis of failure cases revealed that most errors occur in medium-scale car detection under extreme low-light conditions. As shown in Figure 12a,b, insufficient texture details due to poor illumination, coupled with the close proximity of a car and a van, led to clustered point cloud projections with similar color gradients, resulting in the algorithm misidentifying the two distinct vehicles as a single van. In contrast, the baseline model correctly identified the car but missed detecting the nearby van.

These observations validate that while the improved algorithm enhances detection for very large-scale targets, its performance for medium-scale targets closely aligns with the baseline model. This alignment is quantitatively confirmed for car detection precision, as demonstrated in Figure 9.

In summary, the ADF^3^-Net serves as the core component. When used independently, it achieves the highest precision (0.901) and balanced gains, demonstrating its effectiveness in suppressing depth noise. The DTC module, crucial for dual-modal fusion, enhances multi-object detection performance under adverse weather conditions, with increases of 1.16% in mAP50 and 2.27% in mAP50-95. However, it must be paired with a noise-suppression module, as standalone deployment leads to significant precision degradation. When combined with ADF^3^-Net, peak recall (0.820) is attained while maintaining controllable precision. Furthermore, the ADown module significantly reduces computational demands, resulting in a 10.53% reduction in GFLOPs, while preserving the model’s optimal performance. Overall, the improved algorithm significantly enhances detection performance for both small-scale and very large-scale targets, while maintaining moderate performance gains for medium-scale targets.

## 5. Conclusions

This paper proposes a depth-texture coupled multi-modal object detection framework to overcome the limitations of single-modal algorithms in complex autonomous driving scenarios. By designing a depth-texture coupled network, an Adaptive Dimension-Aware Focus Fusion Network (ADF^3^-Net), and an Adown dual-path downsampling module, the framework achieves improved detection accuracy across multiple target scales in challenging traffic environments. Experiments on the KITTI dataset demonstrate a 3.50% increase in mAP50, with particularly significant improvements for large-scale targets as well as small-scale targets such as pedestrians and cyclists. Ablation studies further confirm the complementary contributions of each module. The results highlight the robustness of the framework under occlusion, low illumination, and adverse conditions, providing valuable insights for the advancement of multi-modal fusion in autonomous driving. Future work will focus on improving the perception of dense medium-sized targets under low-light conditions by integrating illumination-adaptive methods to enhance RGB feature clarity, mitigating depth ghosting artifacts through targeted negative sample training, and building a multi-regional benchmark dataset under real adverse weather conditions to validate the robustness of the proposed framework.

## Figures and Tables

**Figure 1 sensors-25-05731-f001:**
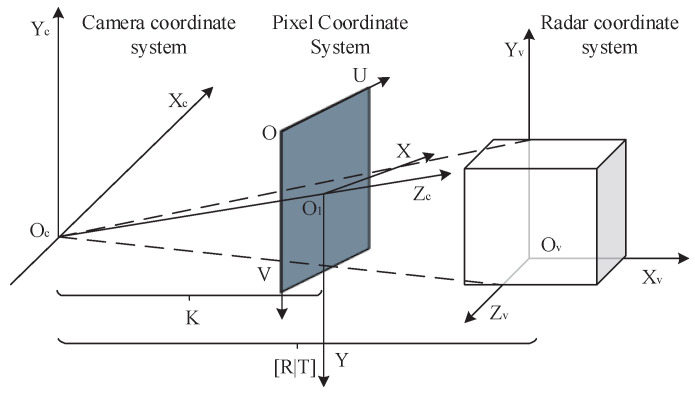
Coordinate system transformations between LiDAR, camera, pixel, and image spaces.

**Figure 2 sensors-25-05731-f002:**
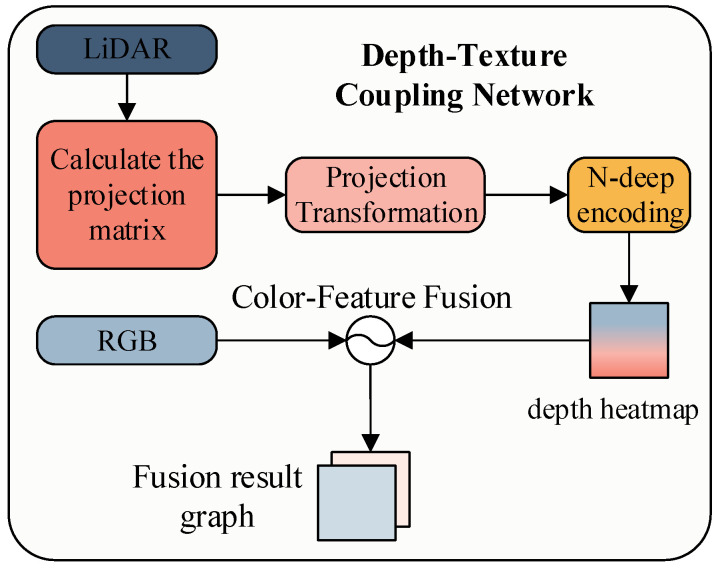
Depth-texture coupling flow chart.

**Figure 3 sensors-25-05731-f003:**
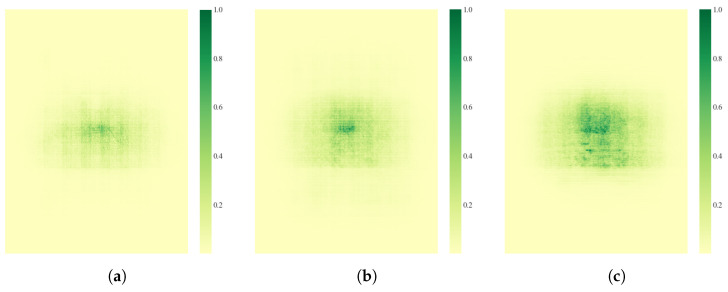
Receptive field comparison under different configurations: (**a**) YOLO11, (**b**) after integrating DTC module, and (**c**) final DTC-YOLO.

**Figure 4 sensors-25-05731-f004:**
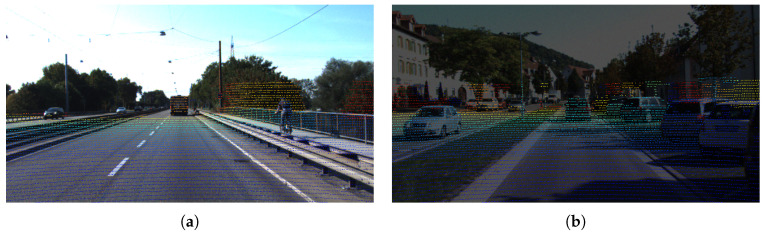
Depth-texture coupling effect diagram. (**a**) and (**b**) are the effects under normal weather and low light respectively.

**Figure 5 sensors-25-05731-f005:**
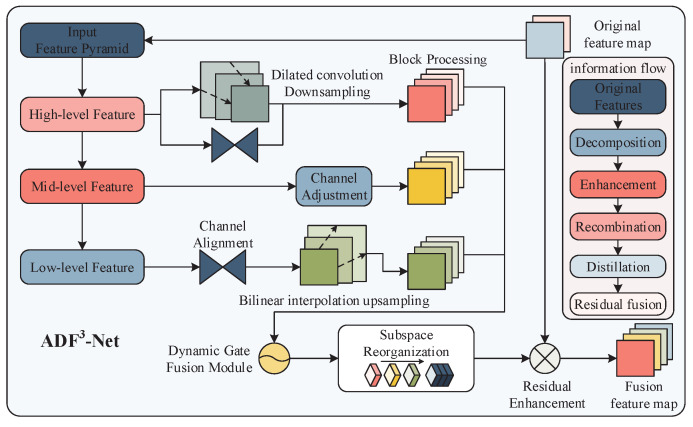
ADF^3^-Net core architecture and information flow.

**Figure 6 sensors-25-05731-f006:**
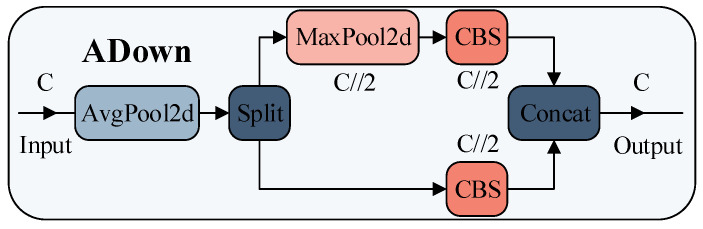
Architecture Diagram of the ADown Downsampling Module.

**Figure 7 sensors-25-05731-f007:**
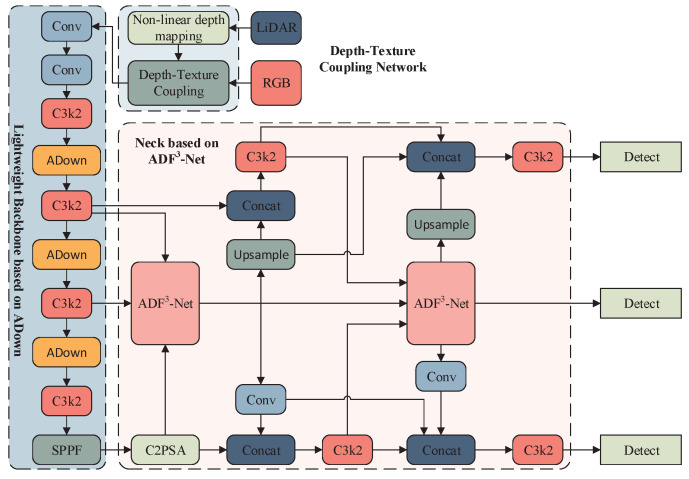
DTC-YOLO network structure.

**Figure 8 sensors-25-05731-f008:**
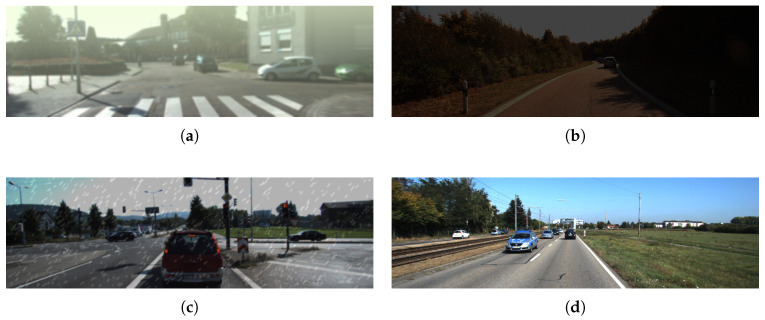
Visualization under different weather conditions: (**a**) foggy, (**b**) low-light, (**c**) rain, and (**d**) clear weather.

**Figure 9 sensors-25-05731-f009:**
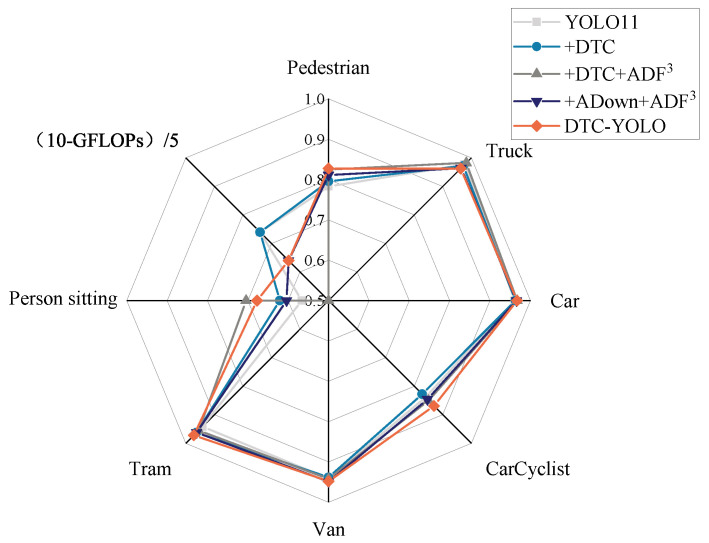
mAP50 and GFLOPs variations across object categories in ablation study.

**Figure 10 sensors-25-05731-f010:**
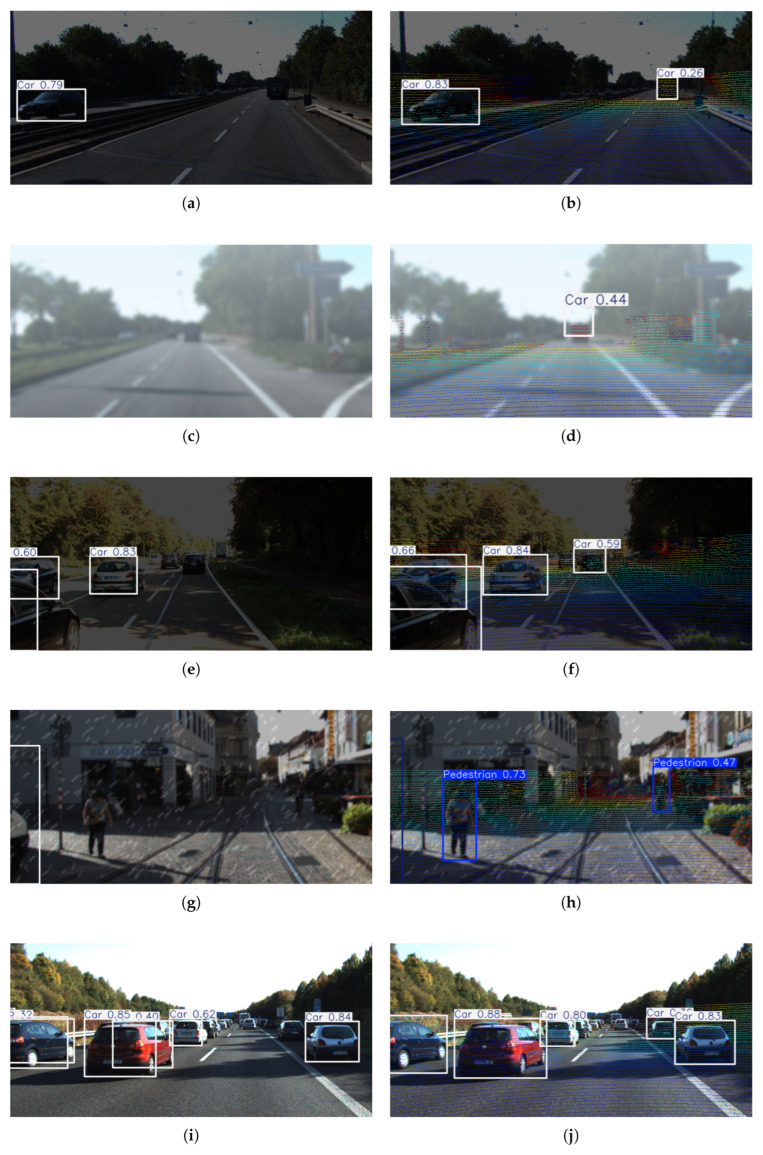
Comparative detection results: YOLO11 (**a**,**c**,**e**,**g**,**i**) vs. DTC-YOLO (**b**,**d**,**f**,**h**,**j**).

**Figure 11 sensors-25-05731-f011:**
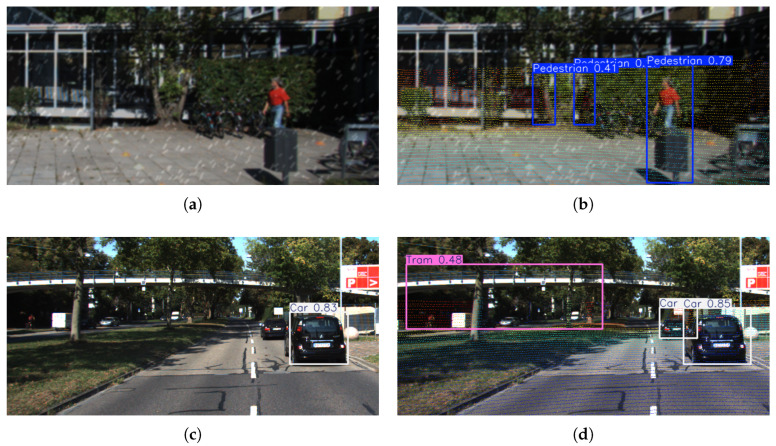
Normal detection (**a**,**c**) vs. depth artifact-contaminated detection (**b**,**d**).

**Figure 12 sensors-25-05731-f012:**
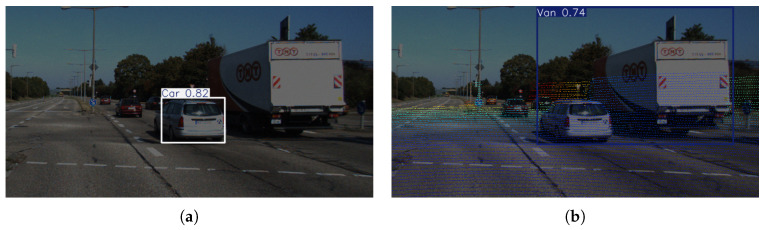
Performance degradation in low-light environments, (**a**) normal detection, (**b**) DTC-YOLO detection failure.

**Table 1 sensors-25-05731-t001:** Confusion Matrix.

	Predicted Positive	Predicted Negative
**Actual Positive**	TP (True Positive)	FN (False Negative)
**Actual Negative**	FP (False Positive)	TN (True Negative)

**Table 2 sensors-25-05731-t002:** Performance comparison of DTC-YOLO and mainstream detectors (KITTI dataset).

Model	mAP50	mAP50-95	GFLOPs
Faster RCNN [41]	0.769	0.449	–
SSD [41]	0.743	0.422	–
MobileNet V3-SSD [42]	0.653	–	12.52
Edge-YOLO [42]	0.726	–	9.97
YOLOv8n	0.896	0.683	8.1
YOLOv10n	0.845	0.630	6.5
YOLOv11n	0.886	0.671	6.8
RT-DETR-r18 [43]	0.859	0.645	57
RT-DETR [43]	0.868	0.649	57
**DTC-YOLO**	0.911	0.693	6.8+

Red indicates best performance; blue denotes second best. “+” denotes potential GFLOPs increase due to unquantified DTC overhead. GFLOPs refers to giga floating point operations per second, equivalent to $10^{9}\;\mathrm{ops}/s$.

**Table 3 sensors-25-05731-t003:** Comparison of DTC-YOLO and other multimodal detectors for pedestrian detection performance.

Multimodal Models [15]	mAP50	mAP50-95
YOLO11n+Normal mapping	0.826	0.488
Faster RNN Based (RGB+LiDAR)	0.809	0.339
Ms-YOLOs (RGB+LiDAR)	0.833	0.453
Ms-YOLOm (RGB+LiDAR)	0.831	0.481
Ms-YOLOL (RGB+LiDAR)	0.809	0.459
MM-Net (RGB+LiDAR)	0.916	0.510
DTC-YOLO	0.842	0.523

Red indicates best performance; blue denotes second best.

**Table 4 sensors-25-05731-t004:** Ablation study results (KITTI simulated weather dataset).

Components				Performance Metrics				
**YOLO11**	**DTC**	**ADown**	**ADF^3^-Net**	**Precision**	**Recall**	**mAP50**	**mAP50–95**	**GFLOPs**
✓				0.866	0.755	0.859	0.617	6.3
✓	✓			0.813	0.814	0.869	0.631	6.3+
✓		✓		0.876	0.792	0.879	0.634	5.3
✓			✓	0.901	0.796	0.882	0.636	7.6
✓	✓	✓		0.875	0.778	0.867	0.624	5.3+
✓		✓	✓	0.848	0.807	0.872	0.631	6.8
✓	✓		✓	0.874	0.820	0.889	0.643	7.6+
✓	✓	✓	✓	0.896	0.803	0.889	0.638	6.8+

Red indicates best performance; blue denotes second best. “+” denotes potential GFLOPs increase due to unquantified DTC overhead. GFLOPs refers to giga floating point operations per second, equivalent to $10^{9}\;\mathrm{ops}/s$.

## Data Availability

The data presented in this study are available in the KITTI Vision Benchmark Suite at [http://www.cvlibs.net/datasets/kitti/, accessed on 30 August 2025], reference number [39]. These data were derived from the following resources available in the public domain: [http://www.cvlibs.net/datasets/kitti/].

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
