# Peer review of "DTC-YOLO: Multimodal Object Detection via Depth-Texture Coupling and Dynamic Gating Optimization"

_sensors, 2025, doi:10.3390/s25185731_

Round 1
Reviewer 1 Report
Comments and Suggestions for Authors
The paper proposes an object detection method that utilizes two data types with different properties: RGB and LiDAR. Using the KITTI dataset, the authors present comparative results against various algorithms, demonstrating the superiority of the proposed method. Additionally, the effectiveness of the proposed algorithm is validated through various ablation studies. The following are several revision points:
1) In Figure 2, an explanation of the N-deep encoding is required. Whether this is a proposed method or adopted from existing work should be clearly stated.
2) Including the original image in the results shown in Figure 7 would be helpful.
3) (Line 265) A description of what the Jet colormap function is should be provided.
4) Figure 4 appears before Figure 3 — the order should be adjusted.
5) The variables in Equation (3) should be explained.
Author Response
1) In Figure 2, an explanation of the N-deep encoding is required. Whether this is a proposed method or adopted from existing work should be clearly stated.
hank you for the reviewer’s comments. There was a naming error in the manuscript. The nonlinear improvement is one of the versions we tried, which also achieved good results. In subsequent experiments, we found that the current improvement approach yielded better overall performance, so we adopted the current version. However, we forgot to update the figure accordingly. We have now corrected the figure.
2) Including the original image in the results shown in Figure 7 would be helpful.
Thank you for the reviewer’s comment. Here, I would like to provide an explanation: Figure 7(a) shows the receptive field image of the baseline model in this paper, which corresponds to the original image.
3) (Line 265) A description of what the Jet colormap function is should be provided.
Thank you for the reviewer’s comment. Regarding the jet mapping function, we have added a more detailed explanation in the main text (lines 240–244).
4) Figure 4 appears before Figure 3 — the order should be adjusted.
Thank you for the reviewer’s comment. We have corrected this ordering error (located on pages 7 and 8 of the main text).
5) The variables in Equation (3) should be explained.
We have added an explanation of the variables in Equation (3) below the equation (lines 240–244).

Reviewer 2 Report
Comments and Suggestions for Authors
- When compared with Yolo model, the specific version should be pointed out, such as s/m/l, and other important indicators such as detection frame rate and parameter quantity should be supplemented.
- Please compare with the detection results of important literatures on multimodal detection in recent 3 years.
The language expression of the paper is not accurate, and there are many grammatical errors.
Author Response
1) When compared with Yolo model, the specific version should be pointed out, such as s/m/l, and other important indicators such as detection frame rate and parameter quantity should be supplemented.
Thank you for the reviewer’s comment. We have added the YOLO version identifiers in Tables 2 and 3 on pages 15 and 16. The YOLO11 version used in this paper is 11n, and all other YOLO versions also use the “n” variant.
2) Please compare with the detection results of important literatures on multimodal detection in recent 3 years.
We have added Table 3 on page 16, which provides a comparative analysis of our algorithm with other multimodal algorithms. The related discussion is presented in the two paragraphs above Table 3 .(lines 512–526).
3)The language expression of the paper is not accurate, and there are many grammatical errors.
Thank you for the reviewer’s comment. We have revised the language and grammar throughout the manuscript, including the Introduction section, lines 246–255, 262–265, 278–285, 335–339, 356–362, 404–421, 558–574, as well as the Conclusion section.

Reviewer 3 Report
Comments and Suggestions for Authors
Reviewer’s comments to the manuscript “DTC-YOLO: Multimodal Object Detection via Depth-Texture Coupling and Dynamic Gating Optimization" (Authors: Wei Xu, Xiaodong Du, Ruochen Li and Lei Xing)
The paper proposes a unique multimodal fusion framework (DTC-YOLO) that integrates RGB and LiDAR data to solve the constraints of single-modal detection in autonomous driving, such as occlusion sensitivity and low environmental adaptation. Both the ADown module for computing efficiency and the ADF³-Net for adaptive feature fusion have strong arguments and experimental support. Ablation investigations and comparisons with cutting-edge models (such as YOLOv8, RT-DETR) show considerable gains in mAP50 (+3.5%) and computational efficiency (10.53% reduction in GFLOPs).
There are some other points to correct or to make the information more exact:
Essential drawbacks.
Remark 1. Depth noise from vegetation/equipment projections leads to false positives (Fig. 11). The proposed mitigation (ADF³-Net) only partially addresses this. The authors do not indicate what steps they envisage in the future to eliminate this problem.
Remark 2. In extremely low light (Fig. 12), the model has trouble detecting medium-scale cars because of crowded point clouds that lead to misidentification. This shows that there are still unsolved issues with feature disambiguation in the presence of sensor noise. How are the authors going to solve these problems in the future to apply the model in real conditions?
Remark 3. The experiments heavily rely on the KITTI dataset, which is supplemented with synthetic adversarial samples (fog, rain, poor light). Real-world testing under harsh conditions is lacking, raising concerns regarding practical applicability. The paper acknowledges this issue, but does not quantify the "physic-consistent deviations" caused by synthetic data.
Technical drawbacks.
Remark 1. Line 14. “CarCyclist”. Should be hyphenated as “Car-Cyclist” or “car-cyclist”.
Author Response
Remark 1. Depth noise from vegetation/equipment projections leads to false positives (Fig. 11). The proposed mitigation (ADF³-Net) only partially addresses this. The authors do not indicate what steps they envisage in the future to eliminate this problem.
Remark 2. In extremely low light (Fig. 12), the model has trouble detecting medium-scale cars because of crowded point clouds that lead to misidentification. This shows that there are still unsolved issues with feature disambiguation in the presence of sensor noise. How are the authors going to solve these problems in the future to apply the model in real conditions?
Thank you for the reviewer’s constructive comments. Regarding the two issues mentioned above, we have added a discussion in the Conclusion section about the remaining limitations of the algorithm and the directions for future improvements (page 20, lines 637–643).
Remark 3. The experiments heavily rely on the KITTI dataset, which is supplemented with synthetic adversarial samples (fog, rain, poor light). Real-world testing under harsh conditions is lacking, raising concerns regarding practical applicability. The paper acknowledges this issue, but does not quantify the "physic-consistent deviations" caused by synthetic data.
Thank you for the reviewer’s comment. Due to funding constraints, we are currently unable to procure equipment to collect our own training data, and publicly available multimodal datasets rarely include data under extreme adverse weather conditions. This makes it difficult for us to quantitatively assess the “physical consistency bias” caused by synthetic data. However, this is indeed a critical step toward practical application and will be a major focus of our future work.
Remark 4. Technical drawbacks.
We have added a discussion in the Conclusion section regarding the technical limitations of the algorithm and the directions for future improvements (page 20, lines 628–643).
Remark 1. Line 14. “CarCyclist”. Should be hyphenated as “Car-Cyclist” or “car-cyclist”.
Thank you for the reviewer’s comment. We have revised all occurrences of “CarCyclist” (three in total) throughout the manuscript to “Car-Cyclist.”

Reviewer 4 Report
Comments and Suggestions for Authors
Title: DTC-YOLO: Multimodal Object Detection via Depth-Texture Coupling and Dynamic Gating Optimization.
In this work DTC-YOLO, a lightweight depth-texture coupled multimodal framework is proposed. Key contributions are: RGB-LiDAR Fusion: Achieves depth-texture coupling via non-linear feature mapping and cross-modal alignment. ADF3-Net: A feature fusion network implementing hierarchical perception, channel decoupling, and spatial adaptation. Its dynamic gated fusion enables adaptive weighted fusion of multidimensional features for enhanced depth-texture interpretation. The work is good, and the authors have presented their work in detail; however, the manuscript currently contains some major and minor corrections (shown below), which should be carefully considered.
Remarks to the Authors: Please see the full comments.
- The Abstract is well; however, it is recommended to rewrite it.
Basically, the Abstract includes background information, the general theme, and the specific research topic. It then describes the problem statements addressed, a brief description of the research methodologies, the main findings, and its results. Finally, it outlines the significance or implications of the findings.
- In literature, various multimodal object detection works have addressed the problems of occlusion sensitivity and adaptation to environmental conditions. Therefore, the contribution of the presented work should be highlighted clearly to present the novelty.
- The introduction section provides a good foundation on the topic of this research based on various works; however, it needs more organization, and some other recent works should be added with much in-depth discussion about them.
- Please check the Abbreviations rules and write them in full term in their first appear only for the entire manuscript.
- The list of contributions mentioned in the Introduction section should be reformulated and shortened to highlight what the current study adds to the existing body of knowledge and demonstrate the significance of the findings rather than stating the methodology of the work.
Besides, in the Abstract, it is mentioned that “Compared to the baseline, DTC-YOLO achieves significant performance gains with only marginal computational cost increase”. However, in the third point of the contributions, it is stated that “……significantly reduces the computational cost”.
It should generally be specifically stated whether the proposed work increases or decreases the computational cost.
- For more organization, the structure of the paper can be added to the end of the introduction section.
Besides, Figure 4 is presented before Figure 3 which leads to reader confusion. And the subfigures of Figure 11 should be defined with their letters in the caption.
- Please cite any information, graph, equation, or data set taken from a previous source with a reliable source, unless it belongs to the authors. Please check this issue for the entire manuscript.
- By using the dynamic gating mechanism, how computational efficiency is maintained. Please give more explanation about this point.
- Please correct the typo error in equation (10) for the subscript (ship). Also check line 393. It is recommended to check this issue for the entire manuscript.
Besides, all variables and symbols used in the entire equations should be defined.
- The adoption of a quantitative comparative analysis of the model’s feature perception ability needs further explanations along with Figure 7, since it gives a good insight into the proposed work performance.
- How have the improvements in both mAP50 (↑ 2.30%) and mAP50-95 (↑ 1.90%) calculated after the integration of the ADF3-Net?
- It is advisable to rewrite the conclusion section into one coherent shorter paragraph. In any scientific research it should contain the proposed work topics and their data, summarize the main points of the work, discuss its importance, and discuss future work. Please review all these points to write a comprehensive shorter conclusion.
The English could be improved to more clearly express the research.
Author Response
1)The Abstract is well; however, it is recommended to rewrite it.
2)Basically, the Abstract includes background information, the general theme, and the specific research topic. It then describes the problem statements addressed, a brief description of the research methodologies, the main findings, and its results. Finally, it outlines the significance or implications of the findings.
3)In literature, various multimodal object detection works have addressed the problems of occlusion sensitivity and adaptation to environmental conditions. Therefore, the contribution of the presented work should be highlighted clearly to present the novelty.
We sincerely appreciate the reviewer’s comments. In response to the three points mentioned above, we have rewritten the abstract, conclusion, and introduction sections, with particular emphasis on revising the contribution part to better highlight the novelty of this study. The newly added literature particularly highlights the challenges of multi-scale object detection, further underscoring the importance of the significant improvements achieved by this study in multi-scale object detection. (Abstract; Introduction, lines 34–43 and 66–86; Conclusion).
4)The introduction section provides a good foundation on the topic of this research based on various works; however, it needs more organization, and some other recent works should be added with much in-depth discussion about them.
5)The list of contributions mentioned in the Introduction section should be reformulated and shortened to highlight what the current study adds to the existing body of knowledge and demonstrate the significance of the findings rather than stating the methodology of the work.
6)For more organization, the structure of the paper can be added to the end of the introduction section.
We sincerely appreciate the reviewer’s comments. In response to the three points mentioned above, we have rewritten the Introduction section, incorporated updated references, and added a brief structural summary at the end of the Introduction (lines 17–86).
7)Please check the Abbreviations rules and write them in full term in their first appear only for the entire manuscript.
Thank you for the reviewer’s comment. We have conducted a thorough check of all abbreviations used throughout the manuscript and corrected multiple instances where full terms were repeatedly replaced by their abbreviations.
8)Besides, in the Abstract, it is mentioned that “Compared to the baseline, DTC-YOLO achieves significant performance gains with only marginal computational cost increase”. However, in the third point of the contributions, it is stated that “……significantly reduces the computational cost”.
It should generally be specifically stated whether the proposed work increases or decreases the computational cost.
It should be clarified that the statement in the abstract, “compared with the baseline, DTC-YOLO achieves significant performance improvement with only a slight increase in computational cost,” refers to the final DTC-YOLO model compared with the original baseline model, where the computational cost increases only marginally.
The third point in the contributions section, “significantly reduces computational cost,” specifically refers to the introduced ADF³-Net module. While this module greatly improves model accuracy, it also substantially increases computational overhead. To address this, we further introduced the Adown module, which effectively reduces the computational cost of ADF³-Net while maintaining its high accuracy. Therefore, overall, the computational cost of the final DTC-YOLO model is still slightly higher than the baseline model but significantly lower than that of the intermediate model using only the ADF³-Net module.
9)Besides, Figure 4 is presented before Figure 3 which leads to reader confusion. And the subfigures of Figure 11 should be defined with their letters in the caption.
Thank you for the reviewer’s comment. We have corrected this ordering error (located on pages 7 and 8 of the main text).
We have modified the caption of Figure 11 to use letters for definitions (page 20).
10)Please cite any information, graph, equation, or data set taken from a previous source with a reliable source, unless it belongs to the authors. Please check this issue for the entire manuscript.
Thank you for the reviewer’s comment. Upon checking the manuscript, we found that some equations indeed lacked citation information. We have now added the appropriate references in the corresponding locations (page 5, lines 200–201).
11)By using the dynamic gating mechanism, how computational efficiency is maintained. Please give more explanation about this point.
Thank you for the reviewer’s comment. We have provided a more detailed explanation in the summary of Section 3.2.2, 2) Adaptive Feature Fusion, on page 9 (lines 300–320).
12)Please correct the typo error in equation (10) for the subscript (ship). Also check line 393. It is recommended to check this issue for the entire manuscript.
13)Besides, all variables and symbols used in the entire equations should be defined.
Thank you for the reviewer’s comment. We have corrected this error and have also checked the manuscript for other equation typographical errors and corrected them. Regarding the symbol definitions, we have added the relevant explanations in the main text (Equations 9 and 10: lines 326–327 and 332–333; symbol definitions on page 7, lines 240–244, and page 9, lines 327–329).
14)The adoption of a quantitative comparative analysis of the model’s feature perception ability needs further explanations along with Figure 7, since it gives a good insight into the proposed work performance.
Thank you for the reviewer’s comment. We have cited Figure 7 in additional appropriate locations in the main text (page 10, lines 335–339).
15)How have the improvements in both mAP50 (↑ 2.30%) and mAP50-95 (↑ 1.90%) calculated after the integration of the ADF3-Net?
Thank you for the reviewer’s reminder. Upon checking, we found that there was indeed a calculation error. The correct values should be mAP50 (↑ (0.889−0.867)/0.867 = 2.54%) and mAP50-95 (↑ (0.638−0.624)/0.624 = 2.24%). In addition, we have reviewed the calculations for other data throughout the manuscript. (page 17, line 559)
16)It is advisable to rewrite the conclusion section into one coherent shorter paragraph. In any scientific research it should contain the proposed work topics and their data, summarize the main points of the work, discuss its importance, and discuss future work. Please review all these points to write a comprehensive shorter conclusion.
Thank you for the reviewer’s comment. We have revised the Conclusion to clearly summarize the main work in a more concise manner and have added discussions on the limitations of the algorithm as well as directions for future work (page 20, lines 627–643).
17) The English could be improved to more clearly express the research.
Thank you for the reviewer’s comment. We have revised the language and grammar throughout the manuscript, including the Introduction section, lines 246–255, 263–265, 278–285, 335–339, 357–361, 405–421, 558–574, as well as the Conclusion section.

Round 2
Reviewer 2 Report
Comments and Suggestions for Authors
1.When comparing with other models in Table 2, please provide specific values for each model's parameters, frame rate, and other indicators.
2.GFLOPs should indicate the unit.
3.There are multiple writing errors in the paper, such as "The main contributions are as follows: RGB-LiDAR Fusion: A" and so on. Please read and revise the entire text carefully.
Comments on the Quality of English LanguageThere are many grammar errors in English, please revise and polish them carefully.
Author Response
1.When comparing with other models in Table 2, please provide specific values for each model's parameters, frame rate, and other indicators.
We thank the reviewer for this valuable comment. We have added the relevant hardware and runtime parameters as suggested (see Table 2 on page 15, lines 488–489). Regarding the frame rate, we would like to kindly clarify that the real-time performance is indeed constrained by the hardware's computational capacity. It is important to note that the primary focus of our current research is on enhancing detection accuracy across various scales, particularly through algorithmic improvements. While optimizing the point cloud projection for higher frame rates is an important direction, it falls slightly outside the immediate scope of this study. Therefore, our comparative analysis primarily centers on key accuracy metrics such as mAP50, which directly reflect our main contributions.
2.GFLOPs should indicate the unit.
Thank you very much for your valuable comments. We have added explanations of GFLOPs and their units in the notes below Tables 3 and 4 (see page 15, lines 488–489; and page 16, lines 531–532).
3.There are multiple writing errors in the paper, such as "The main contributions are as follows: RGB-LiDAR Fusion: A" and so on. Please read and revise the entire text carefully.
Thank you very much for your valuable comments. We have revised this sentence and also checked other expressions in the main text (see page 1, lines 4–6; and page 2, lines 66–67).
4.There are many grammar errors in English, please revise and polish them carefully.
Thank you for the reviewer’s comments. Since the contents of Chapters 1 and 5 have already been revised, in this round we mainly focused on modifying and polishing the sentences in Chapters 3 and 4. The revised and polished paragraphs are as follows: (page 5, lines 190–195 and 198–204; page 6, lines 222–237; page 7, lines 244–259; page 8, lines 273–282; page 9, lines 307–316; page 10, lines 331–341; page 11, lines 367–396; page 12, lines 397–413; page 13, lines 441, 446–453; page 14, lines 471–473, 478–484; page 16, lines 515–516, 520–540, 543–558; page 17, lines 560–570, 574–581; page 18, lines 588–593, 598–609).
All the above modifications are marked in red in the text
Reviewer 3 Report
Comments and Suggestions for Authors
The authors have done significant work to improve the quality of the article. All comments have been answered, and I believe that the work can be published.
Author Response
Once again, I would like to thank the reviewers for their help in improving the paper.
Reviewer 4 Report
Comments and Suggestions for Authors
Most of the comments have been properly addressed and no further comments are needed.
Author Response

(The authors gave the same response as above.)
